# Safety, tolerability, and clinical outcomes of hydroxychloroquine for hospitalized patients with coronavirus 2019 disease

**Michael J. Satlin**[1,2]*, **Parag Goyal**[1,2], **Reed Magleby**[2], **Grace A. Maldarelli**[2], **Khanh Pham**[2], **Maiko Kondo**[1], **Edward J. Schenck**[1,2], **Hanna Rennert**[2,3], **Lars F. Westblade**[1,2,3], **Justin J. Choi**[1,2], **Monika M. Safford**[1,2], **Roy M. Gulick**[1,2]*

**1** Department of Medicine, Weill Cornell Medicine, New York, NY, United States of America, **2** NewYork-Presbyterian Hospital, Weill Cornell Medical Center, New York, NY, United States of America, **3** Department of Pathology and Laboratory Medicine, Weill Cornell Medicine, New York, NY, United States of America

* mjs9012@med.cornell.edu (MJS); rgulick@med.cornell.edu (RMG)

## Abstract

### Background

Severe acute respiratory coronavirus 2 (SARS-CoV-2) has caused a devastating worldwide pandemic. Hydroxychloroquine (HCQ) has *in vitro* activity against SARS-CoV-2, but clinical data supporting HCQ for coronavirus disease 2019 (COVID-19) are limited.

### Methods

This was a retrospective cohort study of hospitalized patients with COVID-19 who received ≥1 dose of HCQ at two New York City hospitals. We measured incident Grade 3 or 4 blood count and liver test abnormalities, ventricular arrhythmias, and vomiting and diarrhea within 10 days after HCQ initiation, and the proportion of patients who completed HCQ therapy. We also describe changes in Sequential Organ Failure Assessment hypoxia scores between baseline and day 10 after HCQ initiation and in-hospital mortality.

### Results

None of the 153 hospitalized patients with COVID-19 who received HCQ developed a sustained ventricular tachyarrhythmia. Incident blood count and liver test abnormalities occurred in <15% of patients and incident vomiting or diarrhea was rare. Eighty-nine percent of patients completed their HCQ course and three patients discontinued therapy because of QT prolongation. Fifty-two percent of patients had improved hypoxia scores 10 days after starting HCQ. Thirty-one percent of patients who were receiving mechanical ventilation at the time of HCQ initiation died during their hospitalization, compared to 18% of patients who were receiving supplemental oxygen but not requiring mechanical ventilation, and 8% of patients who were not requiring supplemental oxygen. Co-administration of azithromycin was not associated with improved outcomes.

**Data Availability Statement:** All relevant data are within the manuscript and its Supporting Information files.

**Funding:** This work was partially supported by the National Center for Advancing Translational Science [UL1 TR002384] at the National Institutes of Health. No additional external funding was received for this study.

**Competing interests:** The authors have declared that no competing interests exist.

## Conclusions

HCQ appears to be reasonably safe and tolerable in most hospitalized patients with COVID-19. However, nearly one-half of patients did not improve with this treatment, highlighting the need to evaluate HCQ and alternate therapies in randomized trials.

## Introduction

Severe acute respiratory coronavirus 2 (SARS-CoV-2) emerged as a cause of lethal illness in December 2019 [1]. In six months, this novel coronavirus caused a worldwide pandemic, infecting over 8 million people and killing over 450,000 people [2]. New York City (NYC) became the epicenter for the disease caused by this virus (COVID-19), with more cases and deaths from this infection than any other city in the world [3]. There are no proven effective therapies for COVID-19 that are available for widespread use. Although the investigational drug remdesivir demonstrated promising preliminary results in one trial [4], its availability is limited and it has yet to be approved by the U.S. Food and Drug Administration (FDA) or European Medicines Agency [5]. Thus, additional data on potential therapies are urgently needed.

   Hydroxychloroquine (HCQ) has *in vitro* activity against SARS-CoV-2 and is an FDA-approved medication for malaria and rheumatologic disorders [6, 7]. However, clinical data supporting its use for COVID-19 are limited to small case series and trials in patients with mild illness, including a study suggesting potential benefit when co-administered with azithromycin [8, 9]. Recent observational studies have not identified an association between HCQ use and improved clinical outcomes in hospitalized patients with COVID-19, although patients who received HCQ were more severely ill than those who did not receive HCQ in these studies [10, 11]. There are also limited data characterizing the safety and tolerability of HCQ for acutely ill patients. Potential serious adverse effects include ventricular arrhythmias due to QT prolongation and hematologic and gastrointestinal toxicity [12–17]. Although large clinical trials to definitively evaluate the efficacy and safety of HCQ for hospitalized patients with COVID-19 have been initiated, results from these trials are not yet available. In the meantime, clinical data describing the use of HCQ for COVID-19 in acutely ill patients are urgently needed. Here we report the safety, tolerability, and clinical outcomes of HCQ for hospitalized patients with COVID-19 during the first three weeks of this outbreak in NYC.

## Methods

### Study population

This retrospective observational study consisted of all patients who were hospitalized at New-York-Presbyterian Hospital/Weill Cornell Medical Center (NYPH/WCMC) and affiliated Lower Manhattan Hospital (NYPH/LMH), had a positive reverse transcriptase-polymerase chain reaction (RT-PCR) test for SARS-CoV-2 between March 05 2020 (date of the first case) and March 25 2020, and had treatment with HCQ initiated during their admission. NYPH/WCMC is a quaternary care referral center and NYPH/LMH is a community hospital. Both hospitals are in Manhattan. Patients who were enrolled in a clinical trial of remdesivir or who were prescribed HCQ chronically were included in the safety and tolerability cohort but excluded from the clinical outcomes cohort (Fig 1).

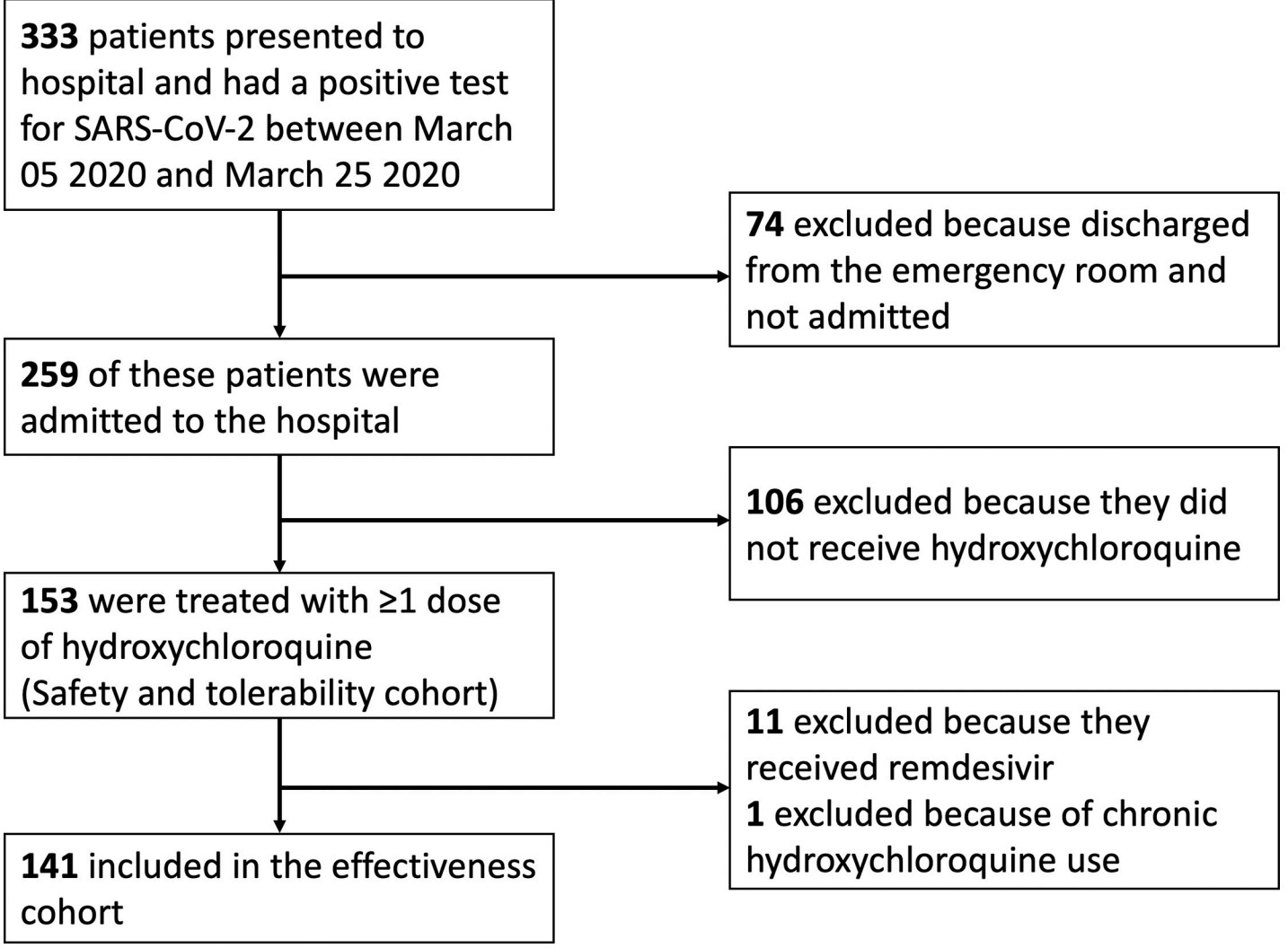

**Fig 1. Study flow diagram.**

Institutional guidance was to offer HCQ to hospitalized patients with COVID-19 who did not have imminent plans for hospital discharge and who had a corrected QT (QTc) interval of <500 msec. After review of pharmacokinetic data [7], a dosage of 600 mg of HCQ every 12 hours for two doses, followed by 400 mg daily for four additional days, was recommended. If patients were able to be discharged prior to completion of this treatment course, HCQ was discontinued upon hospital discharge. Hospitalized patients with COVID-19 who did not receive HCQ generally had mild illness and short inpatient admissions, and thus were not considered to be a suitable control group.

Nasopharyngeal swab specimens were used to assess for SARS-CoV-2 by RT-PCR. This testing was conducted by the New York City Department of Health and Mental Hygiene until March 10 2020 and after March 11 it was conducted at NYPH/WCMC.

### Data collection

Data were retrospectively abstracted manually from the electronic medical record using a quality-controlled protocol and entered into a REDCap database [18]. A random sampling of 10%

of cases to monitor quality yielded high interrater reliability (mean Cohen's kappa of 0.92; [19]). Data included demographics, comorbidities, social characteristics, outpatient medications, presenting symptoms, vital signs (the highest temperature, heart rate, and respiratory rate, and the lowest systolic blood pressure) on the day HCQ was initiated, chest radiography results, administration of HCQ and antimicrobial agents, and outcomes. The study was approved by the Institutional Review Board (#20–03021681) at Weill Cornell Medicine with a waiver of informed consent.

## Safety

We recorded the QTc interval using Bazett's formula for patients who had an electrocardiogram (EKG) performed prior to HCQ initiation [20]. If an EKG was also performed between 1–10 days after HCQ initiation, we compared the baseline QTc interval to the follow-up QTc interval. We also reviewed results of complete blood count and liver tests from the onset of HCQ initiation until 10 days after HCQ initiation. We applied the National Cancer Institute's Common Terminology Criteria for Adverse Events, version 5.0, to assess for incident Grade 3 or 4 neutropenia, lymphopenia, anemia, thrombocytopenia, and aspartate aminotransferase (AST), alanine aminotransferase (ALT), alkaline phosphatase and total bilirubin elevations during this time period [21]. Finally, we recorded the proportion of patients who developed incident kidney failure requiring renal replacement therapy between days 1–10 after HCQ initiation.

## Tolerability

We assessed the proportion of patients who completed their scheduled treatment course, defined as completing the 5-day course or continuing HCQ until the day of discharge. We also reviewed the reasons for discontinuing HCQ. In order to assess incident vomiting or diarrhea, daily inpatient progress notes for each patient were reviewed from the day of HCQ initiation until 10 days after HCQ initiation.

## Clinical outcomes

Our primary clinical outcome was improvement in hypoxia between the day of HCQ initiation and 10 days after HCQ initiation. Hypoxia was graded at baseline, day 5, and day 10 using criteria of the Sequential Organ Failure Assessment (SOFA) Score (S1 Table and S2 Table; [22, 23]). If an arterial blood gas was obtained, we used the lowest partial pressure of oxygen (mm Hg)/fraction of inspired oxygen ($PaO_2/FIO_2$) ratio on the day of assessment. Otherwise, the lowest peripheral capillary oxygen saturation ($SpO_2$)/$FIO_2$ ratio was used. Improvement in the SOFA hypoxia score or discharge between baseline and day 10 after HCQ initiation was considered an improvement in hypoxia; whereas, a worsening score or death was considered worsening of hypoxia.

Secondary outcomes included the need for invasive mechanical ventilation within 10 days after HCQ initiation among patients who were not receiving mechanical ventilation at the start of HCQ therapy. Among patients who had fever (temperature $\geq 38.0$˚C) on the day of HCQ initiation, we assessed the proportion of patients who became afebrile for at least 48 hours within the next 10 days. We also assessed the proportion of patients who survived until hospital discharge and whether patients were discharged to home or to other facilities. The last day of study follow-up was June 19 2020.

## Statistical analysis

Medians and interquartile ranges [IQR] were used to characterize continuous variables and percentages were used for categorical variables. We compared baseline variables between patients with and without improved SOFA hypoxia scores between baseline and 10 days after HCQ initiation, using Fisher's exact or chi-squared tests for categorical variables and the Wilcoxon rank-sum test for continuous variables. A two-sided $P$ value of $\leq$0.05 was used for statistical significance. Variables that were significantly associated with hypoxia improvement were entered into an initial multivariate logistic regression model. Variables were removed from this multivariate model in stepwise fashion until only variables with $P$<0.2 were retained in the final model. Adjusted odds ratios were calculated for each of these variables with 95% confidence intervals (CI). This method was also applied to identify variables that were independently associated with in-hospital mortality. Analyses were conducted using STATA, version 15.0 (StataCorp, College Station, TX).

# Results

## Baseline characteristics

A total of 153 hospitalized patients were treated with HCQ and included in the safety and tolerability cohort (Fig 1). Twelve of these patients were excluded from the clinical outcomes cohort because they also received remdesivir or were prescribed HCQ chronically. The median age of the overall cohort was 62 years (IQR 42–74), 63% were men, one-third were documented as white, and nearly one-third were of Hispanic ethnicity (Table 1). The most common comorbid illnesses were hypertension (50%), obesity (40%), chronic pulmonary disease (32%), and diabetes (24%).

**Table 1. Baseline characteristics of 153 hospitalized patients with COVID-19 who were treated with hydroxychloroquine (HCQ).**

| Patient Characteristics | No. (% of total or IQR) |
|---|---|
| Demographics | |
| Age | 62 (47–74) |
| Female gender | 56 (37) |
| Race | |
| White | 49 (32) |
| Asian | 26 (17) |
| Black | 12 (8) |
| Other | 22 (14) |
| Not specified | 32 (24) |
| Hispanic ethnicity | 44 (29) |
| Comorbid illnesses | |
| Hypertension | 77 (50) |
| Obesity (BMI $\geq$30) | 61 (40) |
| Pulmonary disease | 49 (32) |
| Asthma | 24 (16) |
| COPD | 12 (8) |
| Obstructive sleep apnea | 9 (6) |
| Diabetes | 37 (24) |
| Coronary artery disease | 28 (18) |
| Chronic kidney disease[1] | 17 (11) |

*(Continued)*

**Table 1.** (Continued)

| Patient Characteristics | No. (% of total or IQR) |
|---|---|
| End-stage renal disease | 6 (4) |
| Congestive heart failure | 13 (9) |
| Cerebrovascular disease | 11 (7) |
| Transplant recipient | 8 (5) |
| Solid organ transplant | 6 (4) |
| Bone marrow transplant | 2 (1) |
| Active malignancy | 10 (7) |
| Solid tumor | 6 (4) |
| Hematologic malignancy | 4 (3) |
| Rheumatologic disease | 7 (5) |
| HIV infection | 4 (3) |
| Social characteristics | |
| Current smoker | 5 (3) |
| Former smoker | 35 (23) |
| Recent international travel | 8 (5) |
| Known contact with SARS-CoV-2-infected patient | 20 (13) |
| Healthcare worker | 10 (7) |
| Residence at home prior to hospitalization | 138 (90) |
| Presenting symptoms to hospital | |
| Cough | 128 (84) |
| Fever | 116 (76) |
| Dyspnea | 101 (66) |
| Myalgias | 37 (24) |
| Diarrhea | 36 (24) |
| Nausea or vomiting | 30 (20) |
| Headache | 22 (14) |
| Rhinorrhea | 17 (11) |
| Sore throat | 15 (10) |
| Status on day HCQ initiated | |
| Days from hospitalization until HCQ initiation | 1 (1–2) |
| Highest temperature (˚C) | 38.2 (37.2–38.8) |
| Fever (temperature ≥38.0˚C) | 89 (58) |
| Highest heart rate per minute | 97 (84–107) |
| Highest respiratory rate per minute | 22 (20–28) |
| Tachypnea (respiratory rate ≥22 per minute) | 82 (54) |
| Lowest systolic blood pressure, mm Hg | 103 (93–116) |
| Oxygen support | |
| Ambient air | 28 (18) |
| Low-flow oxygen[2] | 69 (45) |
| Non-invasive mechanical ventilation or high-flow oxygen | 2 (1) |
| Mechanical ventilation | 49 (37) |
| $PaO_2/FIO_2$ ratio | 150 (120–186) |
| Located in an intensive care unit | 54 (35) |
| Laboratory parameters | |
| White blood cell count, in $10^9$ cells/L | 6.1 (4.3–8.3) |
| Absolute lymphocyte count (ALC), in $10^9$ cells/L | 0.9 (0.6–1.2) |
| Lymphopenia (ALC <1x$10^9$ cells/L) | 89 (58) |

(*Continued*)

**Table 1.** (Continued)

| Patient Characteristics | No. (% of total or IQR) |
| --- | --- |
| Hemoglobin, in g/dL | 12.9 (11.7–14.3) |
| Platelet count, $10^9$/L | 182 (138–226) |
| Chest radiography findings | |
| Bilateral infiltrates | 94 (61) |
| Unilateral infiltrates | 30 (20) |
| Pleural effusion | 13 (9) |
| No infiltrate or effusion | 18 (12) |
| QTc interval on electrocardiogram[3] | 442 (420–462) |
| QTc interval ≥470 msec | 20/117 (17%) |
| Antibacterial agents that were co-administered with HCQ[4] | |
| Ceftriaxone | 47 (31) |
| Azithromycin | 27 (18) |
| Doxycycline | 24 (16) |

Variables are expressed as No. (%) or median (IQR), unless otherwise indicated.

Abbreviations: BMI, body mass index; COPD, chronic obstructive pulmonary disease; FIO₂, fraction of inspired oxygen; HIV, human immunodeficiency virus; IQR, interquartile range; PaO₂, partial pressure of oxygen (mm Hg); QTc interval, rate-corrected QT interval using Bazett's formula [20]; SpO₂, peripheral capillary oxygen saturation.

[1]Baseline serum creatinine ≥2 mg/dL.

[2]Includes oxygen via low-flow nasal cannula, face mask, or non-rebreather.

[3]117 patients had an electrocardiogram prior to initiation of HCQ.

[4]Included patients who received the antibacterial agent for ≥3 days and received part of this course at the same time as HCQ initiation.

HCQ was initiated a median of 1 day (IQR 1–2) after hospital admission. Ninety-three percent of patients received the recommended HCQ dosage of 600 mg twice daily for one day, followed by 400 mg daily. The majority of patients had fever and tachypnea on the day of HCQ initiation, 82% required supplemental oxygen, and 36% required invasive mechanical ventilation. The majority of patients had lymphopenia, but white blood cell, hemoglobin, and platelet counts were typically normal. Chest radiography demonstrated bilateral infiltrates in 61% of patients. An EKG was performed prior to HCQ initiation in 76% of patients and the median baseline QTc interval was 442 msec (IQR 420–462). Twenty-seven patients (18%) received ≥3 days of concurrent azithromycin therapy.

## Safety

Forty-seven (40%) of the 117 patients who had a baseline EKG had a follow-up EKG between 1–10 days after HCQ initiation. The median increase in the QTc interval after the initiation of HCQ was 16 msec and 36% of patients had a QTc increase of ≥30 msec. Seven (15%) of these 47 patients had a QTc increase from <500 msec to ≥500 msec and all of these patients received additional medications that prolong the QT interval, including amiodarone (n = 3), azithromycin (n = 3), intravenous ondansetron (n = 3), and anti-psychotic medications (n = 2). Eight-six percent of patients were located on a telemetry unit after HCQ initiation. In total, 13 (9%) patients developed an incident arrhythmia between days 1–10 after HCQ initiation. One patient developed a non-sustained monomorphic ventricular tachycardia that lasted for 15 beats. This patient had a QTc increase from 435 msec to 467 msec after HCQ initiation and was receiving a continuous propofol infusion but did not receive another QT-prolonging

medication. All other arrhythmias were supraventricular tachycardias. No patient developed torsades de pointes, although 9 (6%) died between day 1–10 after HCQ initiation after a do not resuscitate order was implemented and their heart rhythm was not assessed prior to death.

Fifteen percent of patients developed incident Grade 3 anemia and 10% developed incident Grade 3 lymphopenia within the 10 days after HCQ initiation (Table 2). Only 3 patients had a Grade 4 incident blood count abnormality. Grade 3 or 4 AST and ALT increases occurred in 11% and 9% of patients, respectively. Increases in alkaline phosphatase and total bilirubin were rare. Of the 147 patients who were not receiving renal replacement therapy prior to starting HCQ, 19 (13%) required renal replacement therapy within 10 days after HCQ initiation, and all but one of these patients also required vasopressors for hypotension.

## Tolerability

Eight-nine percent of patients completed their HCQ course and the median number of doses was 6 (IQR 4–6). The most common reasons that HCQ was prematurely discontinued were: clinical improvement leading to a decision that additional HCQ was unnecessary (n = 4), QTc prolongation (n = 3), death (n = 3), enrollment into a clinical trial that prohibited HCQ (n = 3), pre-existing thrombocytopenia (n = 2), pre-existing seizure and concern that HCQ might lower the seizure threshold (n = 1), and reason unknown (n = 1). Of 143 patients who did not have vomiting prior to HCQ initiation, 3 (2%) developed vomiting between 1–10 days after HCQ initiation. Of 121 patients who did not have diarrhea at baseline, 7 (6%) developed incident diarrhea.

**Table 2. Incidence of grade 3 and 4 blood count and liver test adverse events within the first 10 days after initiating HCQ therapy in hospitalized patients with COVID-19.**

| Lab value | No. Evaluable[1] | Grade 3: No. (%) | Grade 4: No. (%) |
|---|---|---|---|
| Blood count abnormalities | | | |
| Neutropenia[2] | 136 | 3 (2.2%) | None |
| Lymphopenia[3] | 125 | 13 (10.4%) | 2 (1.6%) |
| Anemia[4] | 134 | 20 (14.9%) | None |
| Thrombocytopenia[5] | 138 | None | 1 (0.7%) |
| Liver test abnormalities | | | |
| AST elevation[6] | 122 | 10 (8.2%) | 3 (2.5%) |
| ALT elevation[6] | 123 | 8 (6.5%) | 2 (1.6%) |
| Alkaline phosphatase elevation[6] | 123 | 2 (1.6%) | None |
| Total bilirubin elevation[7] | 123 | 4 (3.3%) | None |

Grading scale based on the National Cancer Institute's Common Terminology Criteria for Adverse Events (CTCAE), version 5.0.

Abbreviations: AST, aspartate aminotransferase; ALT, alanine aminotransferase.

[1]Assessment for this adverse event was only evaluable in patients who: i) had a baseline laboratory value prior to HCQ initiation and between days 1–10 after initiating HCQ; and ii) did not have a laboratory value that corresponds to a Grade 3 or 4 adverse event on the day of HCQ initiation (blood counts only).

[2]Grades 3 and 4 neutropenia are defined as an absolute neutrophil count of 500–1000 and <500 (x10$^9$ cells/L), respectively.

[3]Grades 3 and 4 lymphopenia are defined as an absolute neutrophil count of 200–500 and <200 (x10$^9$ cells/L), respectively.

[4]Grade 3 anemia is defined as hemoglobin <8 g/dL or need for transfusion. Grade 4 anemia indicates life-threatening consequences where urgent intervention indicated.

[5]Grades 3 and 4 thrombocytopenia are defined as an absolute neutrophil count of 25–50 and <25 (x10$^9$/L), respectively.

[6]Grade 3 AST, ALT, and alkaline phosphatase elevations are defined as an increase >5–20 times the upper limit of normal (ULN) if baseline was normal or >5–20 times baseline if baseline was abnormal. Grade 4 AST and ALT elevations are defined as >20 times ULN if baseline was normal or >20 times baseline if baseline was abnormal.

[7]Grade 3 total bilirubin elevation is defined as an increase >3–10 times the upper limit of normal (ULN) if baseline was normal or >3–10 times baseline if baseline was abnormal. Grade 4 elevation is defined as >10 times ULN if baseline was normal or >10 times baseline if baseline was abnormal.

## Clinical outcomes

Fifty-eight (41%) patients had improvement in their SOFA hypoxia score by day 5 and 73 (52%) had improvement by day 10 (Fig 2). By day 10, 28 (20%) had the same score, and 40 (28%) had a worse hypoxia score. Improvement in SOFA hypoxia score by day 10 occurred in 54% of patients who received HCQ without azithromycin and 41% of patients who received both HCQ and azithromycin (*P* = 0.2). Baseline factors associated with a lack of improvement in the hypoxia score by day 10 in univariate analysis were older age, use of outpatient non-steroidal anti-inflammatory drugs, tachypnea, hypotension, invasive mechanical ventilation, and lymphopenia (Table 3; S3 Table). In multivariate analysis, only age ≥65 (adjusted odds ratio [aOR] 0.43; 95% CI 0.20–0.90; *P* = 0.024) was associated with lack of improvement in hypoxia (Table 3). Concurrent treatment with ≥3 days of azithromycin was not associated with improvement in hypoxia in either univariate (OR 0.63; 95% CI: 0.27–1.49) or multivariate analysis (aOR 0.99; 95% CI: 0.38–2.60).

Of 89 patients who were not receiving mechanical ventilation on the day of HCQ initiation, 23 (26%) were intubated within 10 days after HCQ initiation and only one of these 23 were extubated during this time. Conversely, 12 (23%) of 52 patients who were receiving invasive mechanical ventilation at baseline were extubated within 10 days. Of 78 patients who had fever on the day of HCQ initiation, 62 (80%) became afebrile during the following 10 days, with a median time to defervescence of 3 days (IQR 2–5).

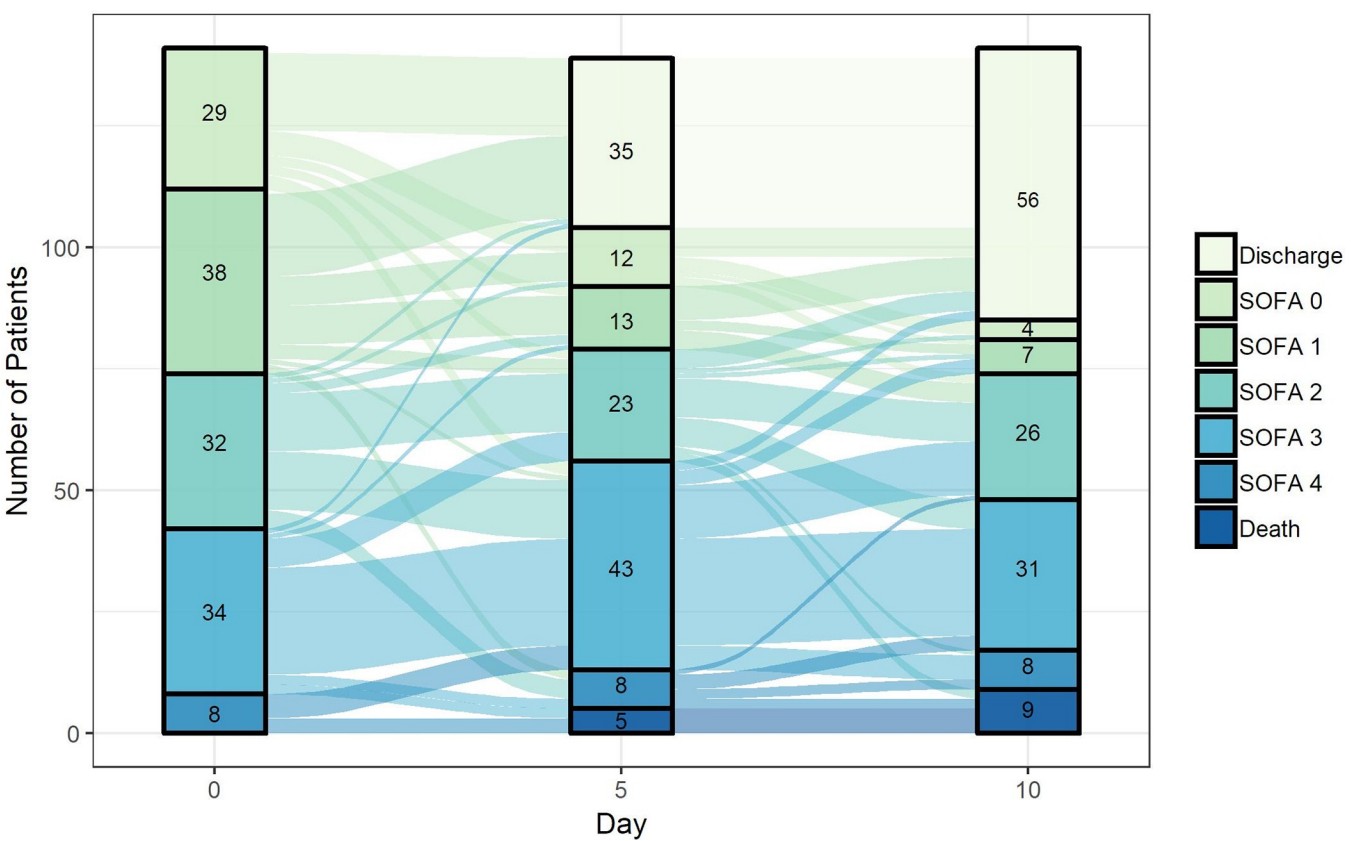

**Fig 2. Sequential Organ Failure Assessment (SOFA) hypoxia scores (0: Least hypoxic; 4: Most hypoxic) at day 0 (day when HCQ initiated), and at days 5 and 10 after HCQ initiation among patients in the clinical outcomes cohort (S1 Table and S2 Table).** The number in each rectangle corresponds to the number of patients who had each score at each study timepoint. Discharge and deaths are reported as their own categories.

**Table 3. Baseline factors associated with improvement in SOFA hypoxia score during the 10 days after treatment with HCQ.**

| | Univariate logistic regression | | Multivariate logistic regression | |
|---|---|---|---|---|
| Variable | Odds ratio (95% CI) | P | Adjusted odds ratio (95% CI) | P |
| Age ≥65 years | 0.32 (0.16–0.64) | 0.001 | 0.43 (0.20–0.90) | 0.024 |
| NSAID use as outpatient | 0.41 (0.18–0.92) | 0.03 | Removed from final model[1] | |
| Tachypnea (respiratory rate ≥22 breaths per min) | 0.43 (0.22–0.84) | 0.014 | 0.52 (0.24–1.12) | 0.095 |
| Hypotension (SBP ≤100 mm Hg) | 0.36 (0.18–0.72) | 0.004 | 0.48 (0.21–1.07) | 0.072 |
| Lymphopenia (<1x10$^9$ cells/μL) | 0.49 (0.24–0.97) | 0.04 | 0.47 (0.22–1.02) | 0.056 |
| Invasive mechanical ventilation | 0.42 (0.21–0.86) | 0.017 | Removed from final model[1] | |
| Azithromycin therapy[2] | 0.63 (0.27–1.49)[3] | 0.29 | Removed from final model[4] | |

Abbreviation: NSAID, non-steroidal anti-inflammatory drug; SBP, systolic blood pressure.

[1]This variable was removed from the final multivariate model because its corresponding P value in the multivariate model was ≥0.2.

[2]Azithromycin used for ≥3 days and administered at the same time that HCQ was initiated.

[3]For comparison, the unadjusted odds ratio of SOFA hypoxia score improvement was 1.31 (95% CI 0.62–2.79) for ceftriaxone therapy and 1.03 (95% CI 0.41–2.60) for doxycycline therapy.

[4]In an alternate multivariate model where azithromycin therapy was retained (with older age, tachypnea, hypotension, and lymphopenia), the adjusted odds ratio for hypoxia improvement with azithromycin therapy was 0.99 (95% CI 0.38–2.60).

By the end of study follow-up, 21% of patients had died during their hospitalization, 77% had been discharged alive, and 2% were still hospitalized. None of the patients who were still hospitalized required supplemental oxygen. Of the patients discharged alive, 69% were discharged to home, 16% to a long-term care facility, 14% to an acute rehabilitation center, and 1% to inpatient hospice. The in-hospital mortality rate was 31% for patients who were receiving mechanical ventilation at the time of HCQ initiation, 18% for patients who were receiving supplemental oxygen but not requiring mechanical ventilation, and 8% for patients who were not requiring supplemental oxygen. Patients who received HCQ and azithromycin had a similar in-hospital mortality rate (22%) as those who received HCQ without azithromycin (21%). In a multivariate model, increasing age (aOR 1.06 per year increase; 95% CI: 1.03–1.10; P = 0.001) and leukocytosis (aOR 5.42; 95% CI: 1.88–15.63; P = 0.002) were independently associated with in-hospital mortality (Table 4). Concurrent azithromycin therapy was not

**Table 4. Baseline factors associated with in-hospital mortality in patients treated with HCQ.**

| | Univariate logistic regression | | Multivariate logistic regression | |
|---|---|---|---|---|
| Variable | Odds ratio (95% CI) | P | Adjusted odds ratio (95% CI) | P |
| Age, per year increase | 1.06 (1.03–1.10) | <0.001 | 1.06 (1.03–1.10) | 0.001 |
| Cerebrovascular disease | 4.24 (1.13–15.78) | 0.031 | 3.52 (0.79–15.64) | 0.098 |
| COPD | 5.30 (1.49–18.13) | 0.010 | Removed from final model[1] | |
| Invasive mechanical ventilation | 2.38 (1.05–5.41) | 0.038 | Removed from final model[1] | |
| Leukocytosis (white blood cell count > 10x10$^9$ cells/μL) | 6.73 (2.53–17.90) | <0.001 | 5.42 (1.88–15.63) | 0.002 |
| Azithromycin therapy[2] | 1.07 (0.39–2.95)[3] | 0.79 | Removed from final model[4] | |

Abbreviation: COPD, chronic obstructive pulmonary disease.

[1]This variable was removed from the final multivariate model because its corresponding P value in the multivariate model was ≥0.2.

[2]Azithromycin used for ≥3 days and administered at the same time that HCQ was initiated.

[3]For comparison, the unadjusted odds ratio of in-hospital mortality was 1.43 (95% CI 0.61–3.34) for ceftriaxone therapy and 0.91 (95% CI 0.28–2.97) for doxycycline therapy.

[4]In an alternate multivariate model where azithromycin therapy was retained (with older age, cerebrovascular disease, and leukocytosis), the adjusted odds ratio for in-hospital mortality with azithromycin therapy was 1.14 (95% CI 0.37–3.49).

associated with a reduction in-hospital mortality in either univariate (OR 1.07; 95% CI: 0.39–2.95) or multivariate analysis (aOR 1.14; 95% CI: 0.37–3.50; Table 4).

## Discussion

There are currently no proven effective therapies for COVID-19 that are available for widespread use. Although remdesivir showed promising preliminary results in one randomized trial [4], results from another trial were less favorable [24], this drug is not currently available for routine use, and it requires intravenous infusion. Thus, data on other potential therapies that are currently available are urgently needed. This report outlines the real-world experience of HCQ therapy for hospitalized patients with COVID-19 at two NYC hospitals. Although no control group was available to conclusively evaluate the efficacy of HCQ, we believe that our findings provide important information to clinicians who are assessing the risk/benefit profile of administering HCQ to acutely ill hospitalized patients with COVID-19.

We found that HCQ was reasonably safe and well-tolerated. Nearly 90% of patients were able to complete their HCQ course and the majority of discontinuations were not related to an adverse event. Incident nausea and vomiting were rare. Although incident Grade 3 or 4 lymphopenia, anemia, and AST elevations occurred in 10–15% of patients, abnormalities in these tests are common in patients with COVID-19 and in critically ill patients [25]. Fewer than 5% of patients developed neutropenia, thrombocytopenia, or elevations in alkaline phosphatase or total bilirubin. Similarly, although renal replacement therapy was required in 13% of patients, COVID-19 commonly causes acute kidney injury in critically ill patients, and almost all of these patients were hypotensive requiring vasopressor support [26, 27].

One major concern of HCQ use in acutely ill patients is the risk of torsades de pointes from QT prolongation [28]. Although supraventricular tachyarrhythmias occurred in nearly 10% of patients, no patient had a documented sustained ventricular tachyarrhythmia or torsades de pointes. However, the majority of patients in this study had an EKG at baseline to assess their QTc interval and three patients discontinued HCQ because of QTc prolongation. Furthermore, among patients with an EKG before and after initiating HCQ, more than one-third had a QTc interval increase of ≥30 msec. Thus, if HCQ is used for hospitalized patients with COVID-19, we believe that obtaining an EKG before and after initiating HCQ is a reasonable strategy to mitigate risk.

The efficacy of HCQ for hospitalized patients with COVID-19 is unknown. Interest in using HCQ for treating COVID-19 arose from its *in vitro* activity against SARS-CoV-2 and its favorable toxicity profile compared to chloroquine [6, 7]. This interest increased after the publication of a small non-randomized study that suggested potential increased virologic clearance of SARS-CoV-2 with the combination of HCQ and azithromycin [8]. This study has been widely criticized for its design [29, 30]. Subsequently, neither two small randomized trials of patients with mild-moderate COVID-19 [16, 31] nor two large observational studies of hospitalized patients identified a clinical benefit of HCQ for COVID-19 [9, 10]. However, results from large randomized trials are still needed to definitely evaluate the clinical efficacy of HCQ in hospitalized patients.

This study did not have a control group of patients who did not receive HCQ because these patients typically had mild disease and were quickly discharged, and thus did not serve as a reasonable comparator group. This lack of a control group precludes any firm conclusions about the efficacy of HCQ for hospitalized patients with COVID-19. However, only approximately one-half of patients had an improvement in their hypoxia within 10 days after initiating HCQ, while over one-quarter had worsening hypoxia or died. These data clarify that even if

HCQ has clinical benefit, which remains uncertain, alternative therapies are desperately needed since a large proportion of patients do not improve with this treatment.

Another notable finding from this study is that patients who were treated with azithromycin and HCQ were not more likely to have improvements in hypoxia at 10 days and were not more likely to survive their hospitalization compared to those who were treated with HCQ only. The lack of additional benefit of azithromycin was identified in both univariate and multivariate analyses. Furthermore, the odds ratio of hypoxia improvement was numerically greater with other commonly used antibiotics used for pneumonia, ceftriaxone and doxycycline, than with azithromycin (Table 3). The number of patients who received concurrent HCQ and azithromycin and had baseline and follow-up EKGs was too small to conduct a meaningful analysis of the additional risk of QT prolongation. However, given that both agents prolong the QT interval [28], the lack of additional clinical benefit of azithromycin argues against routine addition of azithromycin to HCQ in hospitalized patients with COVID-19.

Our study has strengths and limitations. Among its strengths are detailed quantitative assessments of toxicity, tolerability, and clinical outcomes, and extended follow-up for the entire hospitalization in almost all patients. The limitations include an obvious lack of a control group that did not receive HCQ, as well as relying on data that were documented in the medical record. Thus, it is possible that the actual incidence of gastrointestinal adverse events was higher than what was recorded. However, our study team reviewed daily provider notes and our queries identified high interrater reliability for data collection, indicating that relevant data were consistently included in the medical record. Another notable limitation is that our findings cannot be extrapolated to patients who receive HCQ for mild disease or for prophylaxis.

Randomized clinical trials are needed to identify safe and effective treatments for COVID-19, including those that definitively delineate the incidence of adverse effects and efficacy of HCQ in hospitalized patients. In the meantime, we believe that these data add additional preliminary information on the potential risks and benefits of administering HCQ to hospitalized patients with COVID-19.

## Supporting information

**S1 Table. Classification of hypoxia according to Sequential Organ Failure Assessment (SOFA) score criteria[1].**
(DOCX)

**S2 Table. Conversions of supplemental oxygen into $FIO_2$ (fraction of inspired oxygen).**
(DOCX)

**S3 Table. Baseline factors associated with improvement in SOFA hypoxia score during the 10 days after treatment with HCQ.**
(DOCX)

## Author Contributions

**Conceptualization:** Michael J. Satlin, Parag Goyal, Lars F. Westblade, Justin J. Choi, Monika M. Safford, Roy M. Gulick.

**Data curation:** Michael J. Satlin, Parag Goyal, Reed Magleby, Grace A. Maldarelli, Khanh Pham, Maiko Kondo, Edward J. Schenck, Hanna Rennert, Lars F. Westblade, Justin J. Choi, Monika M. Safford.

**Formal analysis:** Michael J. Satlin, Parag Goyal, Reed Magleby, Grace A. Maldarelli, Khanh Pham, Maiko Kondo, Edward J. Schenck, Lars F. Westblade, Justin J. Choi, Monika M. Safford, Roy M. Gulick.

**Investigation:** Michael J. Satlin, Reed Magleby.

**Methodology:** Parag Goyal, Hanna Rennert.

**Project administration:** Michael J. Satlin.

**Resources:** Parag Goyal.

**Writing – original draft:** Michael J. Satlin.

**Writing – review & editing:** Reed Magleby, Grace A. Maldarelli, Khanh Pham, Edward J. Schenck, Hanna Rennert, Lars F. Westblade, Justin J. Choi, Monika M. Safford, Roy M. Gulick.

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
