## [Decision Letter · Decision Letter 0]

1 Jun 2020

PONE-D-20-13406

Safety, Tolerability, and Clinical Outcomes of Hydroxychloroquine for Hospitalized Patients with Coronavirus 2019 Disease

PLOS ONE

Dear Dr. Michael Satlin

Thank you for submitting your manuscript to PLOS ONE. After careful consideration, we feel that it has merit but does not fully meet PLOS ONE’s publication criteria as it currently stands. Therefore, we invite you to submit a revised version of the manuscript that addresses the points raised during the review process.

We look forward to receiving your revised manuscript.

Kind regards,

Muhammad Adrish

Academic Editor

PLOS ONE

Additional Editor Comments:

I have received the comments of the reviewers on your manuscript. Based on reviewer comments, your manuscript warrants publication in PLOS ONE, however, they would like you to revise your paper. Please see reviewer's comments and provide point by point response in your revised version.

Journal Requirements:

2. Please include a copy of Table 3 which you refer to in your text on page 11.

'Funding: This work was partially supported by the National Center for Advancing Translational Science [UL1 TR002384 to Julianne Imperato-McGinley] at the National Institutes of Health'

'The author(s) received no specific funding for this work.'

Reviewers' comments:

Reviewer's Responses to Questions

**Comments to the Author**

1. Is the manuscript technically sound, and do the data support the conclusions?

Reviewer #1: Yes

Reviewer #2: Yes

Reviewer #3: Yes

Reviewer #4: Yes

2. Has the statistical analysis been performed appropriately and rigorously? 

Reviewer #1: Yes

Reviewer #2: Yes

Reviewer #3: Yes

Reviewer #4: Yes

3. Have the authors made all data underlying the findings in their manuscript fully available?

Reviewer #1: Yes

Reviewer #2: Yes

Reviewer #3: Yes

Reviewer #4: Yes

4. Is the manuscript presented in an intelligible fashion and written in standard English?

Reviewer #1: Yes

Reviewer #2: Yes

Reviewer #3: Yes

Reviewer #4: Yes

5. Review Comments to the Author

Reviewer #1: This is a retrospective study on 153 hospitalized patients with COVID-19 who received HCQ. The manuscript is interesting and well written, though limited by an inherent retrospective design and heterogeneity of patients. The main finding is that HCQ is safe and, on average, well tolerated, even though its effectiveness cannot be determined (there is no control group).

-Were other HCQ dosages tested?

-"14% had a QTc increase from <500 msec to ≥500 msec." Did they have an underlying cardiac heart disease? Did they receive other QT-prolonging drugs or other cardiovascular drugs? What about the "One patient [who] developed a non-sustained monomorphic ventricular tachycardia"?

-"Of the 147 patients who were not receiving renal replacement therapy prior to starting HCQ, 19 (13%) required renal replacement therapy within 10 days after HCQ initiation". Were there other plausible causes (drugs, hypotension, septic shock, prior severe renal failure...) for renal function worsening?

-The prognostic role of comorbidities (cardiovascular, renal, respiratory, etc) should be investigated, particularly for deaths and mechanical ventilation. "9 (6%) died between day 1-10"; "Twenty-eight days after HCQ initiation, 19% of patients had died, 13% required mechanical ventilation"

Reviewer #2: This is a well-done study on the use of hydroxychlorochine in COVID patients Several reviews have been written on this field, however only few clinical data are available. This is the principal strength of this paper. Data clearly showed that this drug is well-tolerated and safe in these patients. However, as the Authors pointed out, in absence of a control group, it's impossible to know the clinical advantages of hydroxychlorochine. I suggests to mitigate in the discussion section positive sentences on clinical advantages of this treatment

Reviewer #3: The paper is clear answer for our doubt about the role of hydoxychlorqine in treatment of hospitalized Patients with Coronavirus 2019 and its suspected complications on the different systems of the patients with covid -19

Reviewer #4: In the present study the authors investigated safety, tolerability, and clinical outcomes of hydroxychloroquine (HCQ) for hospitalized patients with coronavirus 2019 disease. The authors concluded that HCQ appears to be reasonably safe and tolerable in most hospitalized patients with COVID-19. However, nearly one-half of patients did not improve with this treatment.

The manuscript is well written and easy to understand. The authors are expert in the field.

However, there are some major concerns and limitations which have to be to addressed:

- What was the exact dosage? 400 mg of HCQ daily for 5 days?

- The authors should re-analysis the findings when patients with HCQ and Azithromycin are excluded; they should show results on patients with only HCQ and HCQ plus Azithromycin. Are there significant differences in outcome?

- The authors should perform multivariate outcome analysis.

- As mentioned a control group that is not receiving HCQ is missing; please state more detailed in the section limitations.

- Please include an discuss the recent study (Observational Study of Hydroxychloroquine in Hospitalized Patients with Covid-19, Geleris et al. NEJM, 05/2020)

- Did you observe a association between hydroxychloroquine use and intubation or death in your population?

- Given the lack of a randomized control group the interpretation of the results are difficult; conclusions made by the authors are difficult to transfer to clinical practice.

6. PLOS authors have the option to publish the peer review history of their article (what does this mean?). If published, this will include your full peer review and any attached files.

Reviewer #1: No

Reviewer #2: Yes: Luciano Agati, MD

Reviewer #3: Yes: Amal Bakry Abdul sattar

Reviewer #4: No

---

## [Author Response · Author response to Decision Letter 0]

30 Jun 2020

Associate Editor:

1) Please include a copy of Table 3 which you refer to in your text on page 11.

Response: We apologize for this error. This was supposed to refer to Supplemental Table 4. We have changed Supplemental Table 4 to Table 3 to include it in the main body of the text. 

2) Thank you for stating the following in the Acknowledgments Section of your manuscript:

'Funding: This work was partially supported by the National Center for Advancing Translational Science [UL1 TR002384 to Julianne Imperato-McGinley] at the National Institutes of Health'

'The author(s) received no specific funding for this work.'

Response: We have removed the funding statement from the manuscript and revised the statement in the online submission form. 

Reviewer #1:

1) This is a retrospective study on 153 hospitalized patients with COVID-19 who received HCQ. The manuscript is interesting and well written, though limited by an inherent retrospective design and heterogeneity of patients. The main finding is that HCQ is safe and, on average, well tolerated, even though its effectiveness cannot be determined (there is no control group).

Response: We thank the reviewer for their favorable comments about our manuscript.

2) Were other HCQ dosages tested?

Response: The vast majority of patients received the recommended dosage outlined in the Methods section. Thus, we were unable to assess other HCQ dosages. We added the following sentence to the Results section: “Ninety-three percent of patients received the recommended HCQ dosage of 600 mg twice daily for one day, followed by 400 mg daily” (lines 174-176, track changes version).

3) "14% had a QTc increase from <500 msec to ≥500 msec." Did they have an underlying cardiac heart disease? Did they receive other QT-prolonging drugs or other cardiovascular drugs? What about the "One patient [who] developed a non-sustained monomorphic ventricular tachycardia"?

Response: We modified this sentence to highlight that these patients received other QT-prolonging medications: “Seven (15%) of these 47 patients had a QTc increase from <500 msec to ≥500 msec and all of these patients received additional medications that prolong the QT interval, including amiodarone (n=3), azithromycin (n=3), intravenous ondansetron (n=3), and anti-psychotic medications (n=2; lines 186-190, track changes version). We also added the following sentence: This patient had a QTc increase from 435 msec to 467 msec after HCQ initiation and was receiving continuous propofol infusion but did not receive other QT-prolonging medications (lines 193-195, track changes version). 

4) "Of the 147 patients who were not receiving renal replacement therapy prior to starting HCQ, 19 (13%) required renal replacement therapy within 10 days after HCQ initiation". Were there other plausible causes (drugs, hypotension, septic shock, prior severe renal failure...) for renal function worsening?

Response: Eighteen of the 19 patients who developed a requirement for renal replacement therapy were hypotensive and on vasopressors. We modified this sentence in the Results section to add: “… and all but one of these patients also required vasopressors for hypotension” (lines 204-205, track changes version).

5) The prognostic role of comorbidities (cardiovascular, renal, respiratory, etc) should be investigated, particularly for deaths and mechanical ventilation. "9 (6%) died between day 1-10"; "Twenty-eight days after HCQ initiation, 19% of patients had died, 13% required mechanical ventilation".

Response: Given that complete follow-up for the entire hospitalization is now available for 98% of the study cohort, we decided to remove the 28-day follow-up and instead characterize the outcome of in-hospital mortality. We constructed univariate and multivariate logistic regression models to assess factors associated with in-hospital mortality, including comorbidities. We revised the Methods section to indicate this change (lines 144-150, track changes version) and the Results section (lines 235-257, track changes version). We have now removed the original Figure 3 and replaced it with Table 4.

Reviewer #2:

1) This is a well-done study on the use of hydroxychloroquine in COVID patients. Several reviews have been written on this field, however only few clinical data are available. This is the principal strength of this paper. Data clearly showed that this drug is well-tolerated and safe in these patients. However, as the Authors pointed out, in absence of a control group, it's impossible to know the clinical advantages of hydroxychlorochine. I suggests to mitigate in the discussion section positive sentences on clinical advantages of this treatment

Response: We thank the reviewer for the favorable comments about our manuscript. We now note in the Discussion section recent studies that cast doubt as to the effectiveness of hydroxychloroquine (references 10, 11, 16; lines 292-295, track changes version). 

Reviewer #3:

1) The paper is clear answer for our doubt about the role of hydoxychlorqine in treatment of hospitalized Patients with Coronavirus 2019 and its suspected complications on the different systems of the patients with covid -19.

Response: We thank the reviewer for their review of our manuscript.

Reviewer #4

1) In the present study the authors investigated safety, tolerability, and clinical outcomes of hydroxychloroquine (HCQ) for hospitalized patients with coronavirus 2019 disease. The authors concluded that HCQ appears to be reasonably safe and tolerable in most hospitalized patients with COVID-19. However, nearly one-half of patients did not improve with this treatment. The manuscript is well written and easy to understand. The authors are expert in the field. However, there are some major concerns and limitations which have to be to addressed:

Response: We thank the reviewer for their comments and thoughtful review of our manuscript.

2) What was the exact dosage? 400 mg of HCQ daily for 5 days

Response: The dosage of HCQ was 600 mg every 12 hours for 2 doses, followed by 400 mg daily for an additional 4 days. This is outlined in the Study Population section of Methods (lines 90-91, clean version). 

3) The authors should re-analysis the findings when patients with HCQ and Azithromycin are excluded; they should show results on patients with only HCQ and HCQ plus Azithromycin. Are there significant differences in outcome?

Response: We thank the reviewer for this excellent suggestion. We added the following sentences to the Clinical Outcomes section of Results: ”Improvement in SOFA hypoxia score by day 10 occurred in 54% of patients who received HCQ without azithromycin and 41% of patients who received both HCQ and azithromycin (P=0.2)” and “Patients who received HCQ and azithromycin had a similar in-hospital mortality rate (22%) as those who received HCQ without azithromycin (21%)” (lines 219-221 and 250-251, track changes version). We also note these results in both univariate and multivariate logistic regression models in Tables 3 and 4 of the revised manuscript.

4) The authors should perform multivariate outcome analysis.

Response: Multivariate models for the outcomes of improvement in hypoxia score at day 10 and in-hospital mortality are now displayed in Tables 3 and 4 of the revised manuscript and are commented upon in the Results section.

5) As mentioned a control group that is not receiving HCQ is missing; please state more detailed in the section limitations.

Response: We outline in the Discussion the important limitation of the lack of a control group who did not receive HCQ in two different paragraphs (lines 298-301, track changes version and lines 327-328, track changes version). 

6) Please include an discuss the recent study (Observational Study of Hydroxychloroquine in Hospitalized Patients with Covid-19, Geleris et al. NEJM, 05/2020).

Response: We thank the reviewer for this suggestion and have added this reference, as well as references 11 and 16. We also added the following sentences to the Discussion section: “Subsequently, neither two small randomized trials of patients with mild-moderate disease [16, 31] nor two large observational studies of hospitalized patients identified a clinical benefit of HCQ for COVID-19 [9, 10]. However, results from large randomized trials are still needed to definitely evaluate the clinical efficacy of HCQ in hospitalized patients (lines 292-297, track changes version).

7) Did you observe a association between hydroxychloroquine use and intubation or death in your population?

Response: Our study did not have a control group of patients who did not receive hydroxychloroquine because essentially all hypoxic patients received hydroxychloroquine during the study period. Thus, we were unable to identify associations between hydroxychloroquine use and intubation or death. 

8) Given the lack of a randomized control group the interpretation of the results are difficult; conclusions made by the authors are difficult to transfer to clinical practice.

Response: We agree with the reviewer’s noted limitation of our manuscript. However, we believe that these data provide clinicians with a detailed analysis of the safety, tolerability, and clinical outcomes of hydroxychloroquine in hospitalized patients with COVID-19.

---

## [Decision Letter · Decision Letter 1]

15 Jul 2020

Safety, Tolerability, and Clinical Outcomes of Hydroxychloroquine for Hospitalized Patients with Coronavirus 2019 Disease

PONE-D-20-13406R1

Dear Dr. Satlin,

We’re pleased to inform you that your manuscript has been judged scientifically suitable for publication and will be formally accepted for publication once it meets all outstanding technical requirements.

Kind regards,

Muhammad Adrish

Academic Editor

PLOS ONE

Additional Editor Comments (optional):

Reviewers' comments:

Reviewer's Responses to Questions

**Comments to the Author**

1. If the authors have adequately addressed your comments raised in a previous round of review and you feel that this manuscript is now acceptable for publication, you may indicate that here to bypass the “Comments to the Author” section, enter your conflict of interest statement in the “Confidential to Editor” section, and submit your "Accept" recommendation.

Reviewer #1: All comments have been addressed

Reviewer #2: All comments have been addressed

Reviewer #4: All comments have been addressed

2. Is the manuscript technically sound, and do the data support the conclusions?

Reviewer #1: Yes

Reviewer #2: Yes

Reviewer #4: Yes

3. Has the statistical analysis been performed appropriately and rigorously? 

Reviewer #1: Yes

Reviewer #2: Yes

Reviewer #4: Yes

4. Have the authors made all data underlying the findings in their manuscript fully available?

Reviewer #1: Yes

Reviewer #2: Yes

Reviewer #4: (No Response)

5. Is the manuscript presented in an intelligible fashion and written in standard English?

Reviewer #1: Yes

Reviewer #2: Yes

Reviewer #4: (No Response)

6. Review Comments to the Author

Reviewer #1: The Authors addressed all comments and improved the manuscript. I have no more questions / comments.

Reviewer #2: The Authors correctly reply to my suggestions, more recent articles on the same field were cited . I don't have further questions

Reviewer #4: The authors addressed all requests and comments. The present work might fulfill criteria for publication.

7. PLOS authors have the option to publish the peer review history of their article (what does this mean?). If published, this will include your full peer review and any attached files.

Reviewer #1: No

Reviewer #2: **Yes: **luciano agati, MD

Reviewer #4: No

---

## [Editor Report · Acceptance letter]

16 Jul 2020

PONE-D-20-13406R1 

Safety, Tolerability, and Clinical Outcomes of Hydroxychloroquine for Hospitalized Patients with Coronavirus 2019 Disease 

Dear Dr. Satlin:

I'm pleased to inform you that your manuscript has been deemed suitable for publication in PLOS ONE. Congratulations! Your manuscript is now with our production department. 

Kind regards, 

on behalf of

Dr. Muhammad Adrish 

Academic Editor

PLOS ONE